# Prevalence and Distribution of Hard Ticks and Their Associated Risk Factors in Sheep and Goats from Four Agro-Climatic Zones of Khyber Pakhtunkhwa (KPK), Pakistan

**DOI:** 10.3390/ijerph191811759

**Published:** 2022-09-17

**Authors:** Zubaria Iqbal, Amjad Rashid Kayani, Ali Akhter, Mazhar Qayyum

**Affiliations:** Department of Zoology, Wildlife and Fisheries, Faculty of Sciences, PMAS-Arid Agriculture Rawalpindi, Murree Road, Rawalpindi 46300, Pakistan

**Keywords:** epidemiology, hard ticks, agro-climatic zones, Khyber Pakhtunkhwa (KPK), Pakistan

## Abstract

Hard ticks are blood-sucking ectoparasites infesting all types of domestic ruminants throughout the world and serve as vectors in the dissemination of a wide variety of pathogens. Sheep and goat farming is a vital economic source for resource-poor farming communities in the Khyber Pakhtunkhwa (KPK) province of Pakistan. **Aim:** The aim of present study is to study the epidemiological profile of ticks in various agro-climatic zones of KPK. **Materials and Methods:** A total of 1500 (882 sheep and 618 goats) of all age groups and sexes were examined for the presence of ticks belonging to six districts in four agro-climatic zones of KPK province, Pakistan. Non-probability sampling was undertaken based on animal hosts’ selection, collection of hard ticks, and epidemiological parameters. Ticks collected from sheep and goats were identified phenotypically using standard keys. **Results:** The results revealed that *Rhipicephalus microplus* (36.2%) was the predominant tick species followed by *Hyalomma anatolicum* (25.2%), *Hyalomma trancatum* (18.1%), *Hyalomma aegyptium* (11.3%)*, Hyalomma asiaticum* (6.9%), and *Haemephysalis bispinosa* (2.4%). Different epidemiological parameters showed that weather, humidity, and host age negatively affect (*p* < 0.05) tick load while temperatures (Minimum and Maximum), sunshine (hrs), and precipitation positively affected (*p* < 0.05) tick load. Host sex only showed a positive association with tick load (*p* > 0.05). The highest value of tick diversity index (H) 0.36748 was noted for *Hyalomma anatolicum* as this tick species was commonly distributed in three agro-climatic zones except in the Suleiman Piedmont zone, while *Haemaphysalis bispinosa* has the lowest diversity index (0.0334) as it was only found in the Central Valley Plains zone of KPK. The Shannon diversity index of tick diversity was highest at Barikot while the lowest index value was at Kabal (2.407). Similarly, a higher Simpson’s diversity index was at Barikot (0.9554) and the lowest hard index was noted at Takht bai (2.874). The dominance index showed that the tick species were more dominant at Takht bai (0.1251), while the lowest dominance was at Barikot (0.04465). **Conclusions**: It has been concluded that tick diversity and distribution, abundance, reproduction, development, and survival depend on prevailing climatic conditions. The present study would not only help to identify the tick species but also facilitate in devising effective control measures to minimize the transmission pathogens in sheep and goats being reared in the various agro-climatic zones of KPK province, Pakistan.

## 1. Introduction

Sheep and goat farming is an essential component of animal production systems in Pakistan and contributes significantly to the development of the livestock sector. Asian countries have the largest sheep and goat populations as it is 35.2% in India, 29.3% in China, and 12.0% in Pakistan [1]. Small ruminants play a vital role in poverty alleviation in resource-poor farming communities [2,3,4]. In Pakistan, livestock production potential remains vibrant and signifies the foremost asset among resource-poor farmers [5]. Globally, hard ticks have an important role in decreasing the production potential in the livestock sector [6]. Ticks are extremely deleterious for livestock as they inflict a huge financial impact, especially in sub-tropical and tropical regions [7,8]. It has been estimated that USD 7.0 billion annual losses are attributed to hard tick infestations in small and large ruminants [9]. *Rhipicephalus microplus* has inflicted heavy economic losses (USD 22–30 billion) in the livestock sector [10].

Ticks are blood-sucking ectoparasites infesting all types of domesticated ruminants [11,12]. They serve as vectors for the transmission of zoonotic infections caused by various pathogens. Previous studies indicated that the emergence and reemergence of various tick-borne infections cause serious threats to general public health viz., Anaplasmosis, Cremian Congo Hemorrhagic Fever (CCHF), Bebesiosis, Borreliosis, Rickettsiosis, Relapsing fever (RF), tularemia, looping ill, tick-borne encephalitis (TBE), African tick bite fever (ATBF), Powassan encephalitis (PE), Human Monocytic Ehrlichiosis (HME), Mediterranean spotted fever (MSF), and Q fever, Tularemia [13,14,15]. In Pakistan, 94.4 % of ticks were infected with tick-borne pathogens [16]. It was further elaborated that 43.4% of ticks were carrying one pathogen followed by two (38.9%), three (14.5%), four (2.3%), and five (0.9%). The predominant pathogens found in different tick species were *Babesia* spp., *Theileria* spp., *Ehrlichia* spp., and *Anaplasma marginale*. The predominant pathogens found were *Babesia* sp. and *Theileria* spp., *Ehrlichia* spp., and *Anaplasma marginale* in different tick species. *Hyalomma anatolicum*, *Rhipicephalus microplus, Hyalomma hussaini*, and *Rhipicephalus annulatus* have zoonotic significance as they are considered to be involved in the transmission of Crimean Congo hemorrhagic viral infection in Pakistan.

The livestock sector is a vital agro-economic component in Pakistan. Ticks also directly or indirectly cause substantial monetary losses in the livestock sector of Pakistan. Different tick species have been reported from various agro-climatic zones (ACZs) of Pakistan [17,18,19]. *Hyalomma anatolicum, Hyalomma marginatum, Hyalomma truncatum, Hyalomma dromedarii, Hyalomma rufipes, Rhipicephalus sanguineus, Rhipicephalus appendiculatus, Boophilus microplus,* and *Boophilus decolaratus* tick species were recorded in different agroecological zones of Punjab [20]. 

Previous surveillance studies on tick diversity were representative of different areas which do not truly demonstrate the target areas. In KPK province, a detailed epidemiological investigation has not been carried out in different agro-climatic zones. Therefore, epidemiological data accessible from other regions cannot be used to devise effective control measures to target tick populations in the proposed areas. Recent climate changes in Pakistan can also impose serious repercussions on the epidemiological pattern in different ecological sittings. It is essential to determine the epidemiological profile of ticks in various agro-climatic zones of KPK, where sheep and goat farming are key initiatives to uplift the socio-economic conditions of resource-poor farmers. Furthermore, the prevalence and distribution of ticks in various agro-climatic zones of KPK province will provide potential vectors for dissemination of tick-borne pathogens in sheep and goats. The present study would also help to implement strict management guidelines and recommend accurate use of acaricide treatments to reduce the tick densities in KPK. The present study aimed to determine the epidemiological profile of hard ticks prevalent in small ruminants being reared in various agro-climatic zones of Khyber Pakhtunkhwa (KPK), Pakistan.

## 2. Material and Methods

### 2.1. Agro-Climatic Zones of the Area

The present study was carried out in four agro-climatic zones of KPK province of Pakistan. The study sites were selected based on the following criteria: the area’s security concern, accessibility by roads, availability of sheep and goats at farms/flocks, and cooperation offered by the farmers. Different small ruminant herds were visited on a monthly basis. Six districts fall within the jurisdiction of four agro-climatic zones, namely: I. Sub-humid Eastern Mountains (Mansehra) with 200 mm monthly rainfall in summer and 35–50 mm in winter; II. Higher Northern Mountains (Mingora) characterized by a mean monthly rainfall of 235 mm in summer and 116 mm in winter; III. Central Valley Plains (Peshawar and Mardan), the climate is semi-arid with 20–30 mm mean monthly rainfall; and IV. Suleiman Piedmont (Bannu and Dera Ismail Khan), the climate of this zone is arid and hot and mean monthly rainfall is less than 15 mm. The geographic location and coordinates of the selected sites are provided in Figure 1 and Table 1, respectively. The average distance between each site is about 50–100 km. Topographically, study sites were categorized into hilly, semi hilly, and plain areas. 

### 2.2. Study Design

Non-probability sampling was undertaken based on animal host selection, collection of hard ticks, and epidemiological parameters. Purposive sampling was used based on the following principles. We selected 1 male and 3 female hosts from each farm at each visit. The age of the animals was also considered as more than 50% of animals were from the 1–2 year age group and 20% from the >1 year and above 2 years.

### 2.3. Collection of Ticks

Ticks were collected during May 2017 to April 2018 from 1500 animals (882 sheep and 618 goats). For the epidemiological investigation a detailed questionnaire was designed with information on host gender age, tick species, socio-economic settings, and management practices of herd owners. Sheep and goats’ age estimation was completed by examining the number of incisor teeth present on the lower jaw as described by [21]. Ticks were searched by examining all the body parts and picked up with the help of rubber-coated forceps without damaging mouthparts. Ticks were transferred to McCartney sample collection bottles containing 70% ethanol and 5% glycerin for preservation. 

### 2.4. Identification of Ticks

Ticks were processed for identification and examined under a stereomicroscope (Olympus-CH20 BIM, Japan). Tick identification was carried out to the species level using standard keys and descriptions [22,23].

### 2.5. Meteorological Data

Meteorological data (Minimum and Maximum Temperatures, Sunshine, Precipitation, Humidity, and Rate of Evaporation) was obtained courtesy of the Pakistan Meteorological Department, Pakistan, National Weather Forecasting Centre, H-8, Islamabad. 

### 2.6. Statistical Analysis

Data were analyzed using Statistical Package (SPSS 20 IBM, USA). The data were entered and managed in Microsoft Excel sheet. The SPSS 20 version software program was used for data analysis including minimum, maximum, averages, and correlation analysis. The data of prevalence and categorical variables viz., host’s age, sex, and sites were correlated using the Chi-Square test (χ^2^). The *p*-value was calculated separately for each variable and a *p* value less than 5 % (*p* < 0.05) was taken as statistically significant. Tick diversity indices, the Shannon–Wiener index (H′) and Evenness index (E), were calculated using Past 4.03 version.

## 3. Results

### 3.1. Overall Prevalence of Tick Species

In the current study, six species were identified viz., *Rhiphicephalus microplus, Hyalomma trancatum, Hyalomma anatolicum, Hyalomma aegyptium, Hyalomma asiaticum,* and *Haemaphysalis bispinosa*, from various districts of KPK (Table 2). The results revealed that Rhipicephalus microplus (36.2%) was the predominant tick species followed by *Hyalomma anatolicum* (25.2%), *Hyalomma trancatum* (18.1%), *Hyalomma aegyptium* (11.3%), *Hyalomma asiaticum* (6.9%), and *Haemephysalis bispinosa* (2.4%).

### 3.2. Correlation Matrix between Epidemiological Parameters 

The data on the correlation coefficient (r) calculated between different parameters affecting tick prevalence is presented in Table 3. Weather, humidity, and host age negatively affect (*p* < 0.05) tick load while temperatures (Minimum and Maximum), sunshine (hrs), and precipitation positively affect (*p* < 0.05) tick load. Host sex only showed a positive association with tick load (*p* > 0.05).

### 3.3. Tick Prevalence in Agro-Climatic Zones 

All six ticks were prevalent in Central Valley Plains with varied prevalence rates (Table 4). In Suleiman Piedmont, three species were recorded, namely: *Rhipicephalus microplus* (63.07%), *Hyalomma trancatum* (28.07%), and *Hyalomma asiaticum* (8.84%), whereas in Sub-humid Eastern Mountains and Higher Northern Mountains only *Hyalomma anatolicum* was the predominant tick species (100% in both zones) with a prevalence rate.

### 3.4. Distribution of Hard Ticks in Different Sub-Sites 

There was significant difference (*p* < 0.05) related to frequency distribution of hard ticks with reference sampling sites, land topography, host gender, and age groups in sheep and goats (Table 5). 

### 3.5. Ticks Diversity Index

The highest value of the tick diversity index (H) 0.36748 was noted for *Hyalomma anatolicum* as this tick species was commonly distributed in three agro-climatic zones except in Suleiman Piedmont zone, while *Haemaphysalis bispinosa* has the lowest diversity index (0.0334) as it was only found in the Central Valley Plains zone of KPK. Diversity indices for other tick species were found for *Rhipicephalus microplus* (0.36665), *Hyalomma asiaticum* (0.2985), *Hyalomma truncatum* (0.24195), and *Hyalomma aegyptium* (0.05756) as shown in Table 6. *Hyalomma anatolicum* was commonly found in three agro-climatic zones except in Suleiman Piedmont zone, while *Haemaphysalis bispinosa* was only found in Central Valley Plains of KPK province.

### 3.6. Diversity Indices at 12 Different Sampling Sites

There was no significant difference among sampling sub-sites (*p* > 0.05). The diversity indices are presented in Table 7. The evenness indices of hard tick communities were greater in Nurar (0.85) followed by Kabal (0.79), Topan wala (0.58), Barikot and Chankari (0.53), Shorkot (0.51), Darmangi (0.50), Rashakai (0.48), Tajik khula, Sufaida and Takht bai (0.47), and Dhangri (0.44). The Shannon diversity index of hard tick communities was highest in Barikot followed by Dhangri (3.443), Topan wala (3.203), Rashakai (3.162), Sufaida (3.147), Tajik hula (3.133), Darmangi (3.036), Chankari (2.957), Nurar (2.88), and Takht bai (2.874). On the other hand, the lowest diversity index was at Kabal (2.407). Simpson’s diversity index indicated that higher tick diversity was at Barikot (0.9554) followed by Nurar (0.9372), Topan wala, Taji khula, sufaida, Rashakai, and Dhangri (0.91), followed by Chankari, Kabal, and Darmangi by (0.8931), followed by Shorkot (0.8824) and the lowest hard tick diversity was noted at Takht bai (2.874).

### 3.7. Dominance Index

The dominance index showed that the tick species were more dominant in Takht bai (0.1251) followed by Shorkot (0.1176), Chankari, Darmangi and Kabal (0.1074), Topan wala, Tajik hula, Sufaida, Rashakai, Dhangri (0.08112), and Nurar (0.06278), while the lowest dominance was found in Barikot (0.04465). The dominance of different tick species is shown in Figure 2.

### 3.8. Equitability

The equitability index indicated that species were evenly distributed at Nurar (0.9468) followed by Kabal (0.912), Barikot (0.862), Topan wala (0.8569), Chankari (0.8252), Darmangi, Dhangri, Rashakai, Shorkot (0.81), and Sufaida and Tajik hula (0.8087), and the lowest was found in Takht bai (0.796).

## 4. Discussion

In this study, six tick species *Rhiphicephalus microplus, Hyalomma trancatum, Hyalomma anatolicum, Hyalomma aegyptium, Hyalomma asiaticum*, and *Haemaphysalis bispinosa*, were identified from four agro-climatic zones of KPK. Similar tick species except *Haemaphysalis bispinosa* were also reported by [3,17] in KPK; [24] in semi-dry and dry areas of the Punjab province, Pakistan, and in southern Punjab, Pakistan [3]. Similar tick fauna may be due to extensive transportation and movements of animals from one zone to another in search of new grazing land. Additionally, the tick distribution pattern in various climatic zones of KPK might also be due to prevailing climatic conditions and animal husbandry management practices in the study area. It is reported that macro- and micro-climatic settings of an area are the key critical factor to control the distribution pattern of hard ticks [25]. Our results showed that tick infestation rates were higher in hilly, semi-hilly, and plain areas as compared to other areas of Pakistan [26,27,28]. These results are in agreement with [29] that favorable environmental conditions might facilitate the development and survival of different tick species. Another plausible explanation is the type of vegetation and presence of coarse pastures in a particular habitat. Furthermore, the local micro-environment, pasture height, and humidity are the crucial factors that facilitate the development and survival of different tick stages [30]. In the present study *Rhipicephalus microplus* was recorded in hot semi-arid climates which is not in agreement with findings of [31]. In this study, they mentioned that the high prevalence of *R. microplus* usually occurs in areas receiving more annual precipitation. On the other hand, the present results are consistent with [32] who reported *R. microplus* in semi-arid and arid zones of Punjab, Pakistan. In Western African countries, this species is considered an invasive tick species that might be involved in the change in the local tick fauna [33]. The nomadic herders in KPK may facilitate the distribution of *Rhiphicephalus microplus* in new areas due to extensive movement of herds in search of new grazing lands. *Hyalomma analoticum* is a predominant tick recovered from Mardan, Mansehra, and Mingora districts which indicates its wide prevalence in many parts of KPK province. This finding is in agreement with [34] in Sudan and Ref. [35] in Afghanistan, Ref. [36] in Qazvin province (Iran), Ref. [37] in southeastern (Iran), and Ref. [38] in Isfahan province (Iran), according to them the genera *Hyalomma* and *Rhipicephalus* have adapted in Middle East climatic conditions with cold winter and dry summer seasons. In the present study, *Hyalomma asiaticum* was reported for the first time in small ruminants in KPK which is agreement with [39,40] in Iran.

*Haemaphysalis bispinosa* was reported for the first time in Central Valley Plains of KPK province. Previously, this tick species was also reported by various authors throughout the world with varied prevalence rates [4,41,42,43]. Seasonal climatic conditions are one of the important factors that help to determine the distribution of ticks from all over the world. It is observed that the prevailing rainy summer could facilitate the reproduction, development, and growth of different tick species in various agro-ecological zones (AEZs) of Pakistan [4,43,44,45]. In KPK, weather conditions are generally hot and humid that facilitates the growth and survival of ticks during the hot summer season. It has also been observed that tick genera, namely *Rhipicephalus, Hyalomma*, and *Haemaphysalis* have become adapted in the northwestern part of Pakistan characterized by dry summer and cold winter weather conditions. This has indicated that these three genera are widely distributed in various agro-ecological zones (AEZs) of Pakistan. Similar findings were also reported by [7,9,24,45,46], according to them the tick prevalence rate was high during the wet monsoon as compared to the dry summer season. In Khyber Pakhtunkhwa (KPK), the livestock farmers move their animal herds during the summer season towards northern hilly areas where plenty of lush green grazing lands are available to their animals. This migration pattern of nomads might predispose their animal to tick infestations. It has been observed that hilly areas of northern KPK like the Swat district offer a greater chance to build up the load on sheep and goats. The seasonal movement of animals from one area to other areas not only exposes the animals to the acquisition of tick infestation but also favors the dissemination of ticks to new areas where they had never previously been reported. This animal movement practice may not only change the local tick fauna but also introduce new tick-borne diseases (TBDs). A similar observation was also made by [47]. According to them, regular trans-boundary movement of animal herds is used to explore the seasonal availability of pastures for their animals. Such movements unfortunately have resulted in the spread of new ticks along with their pathogens into new areas in KPK, where they were not reported in the past. It is found that all the tick species recovered have the potential to transmit deadly pathogens *viz., Rickettsia, Babesia, Theileria,* and Congo virus in sheep and goats in the studied areas. These findings are consistent with different researchers who have reported that the tick species are the major source of tick-associated pathogens [4,48,49,50]. Therefore, the role of the reported ticks in the transmission of zoonotic pathogens cannot be ignored in study areas. 

## 5. Conclusions

The present investigation provides comprehensive information regarding the epidemiology of hard ticks in various agro-climatic zones of Khyber Pukhtunkhaw (KPK). It was observed that due to regular movement of animal herds, to explore new grazing land, the introduction new tick species in the different zones may occur. Previous studies showed that reported tick species are involved in the transmission of deleterious pathogens of zoonotic impotence. Therefore, this investigation will be helpful to trace the dissemination of TBDs in the animals of these zones. The present study was conducted for the first time to determine the diversity indices of ticks in the selected districts of KPK and will pave the way to investigate tick diversity patterns in various agro-climatic zones of Pakistan.

## Figures and Tables

**Figure 1 ijerph-19-11759-f001:**
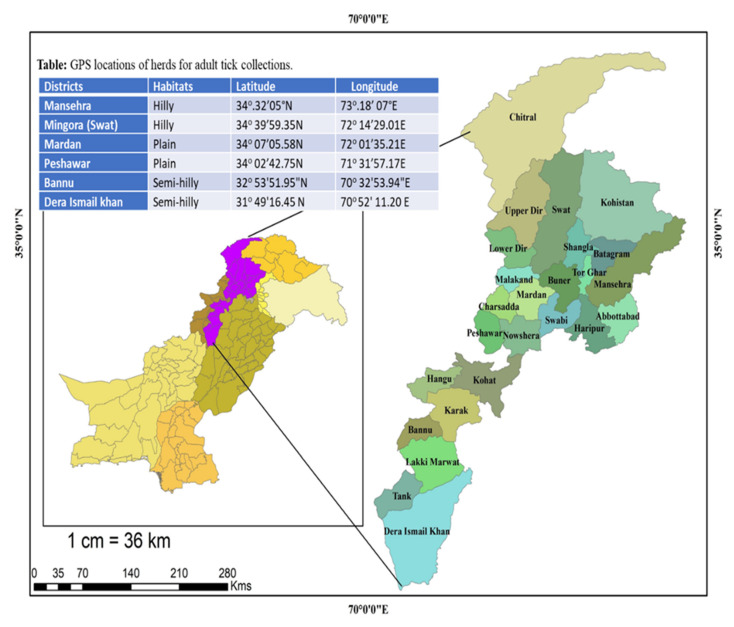
Map depicting the study sites located in Khyber Pakhtunkhwa (KPK), Pakistan.

**Figure 2 ijerph-19-11759-f002:**
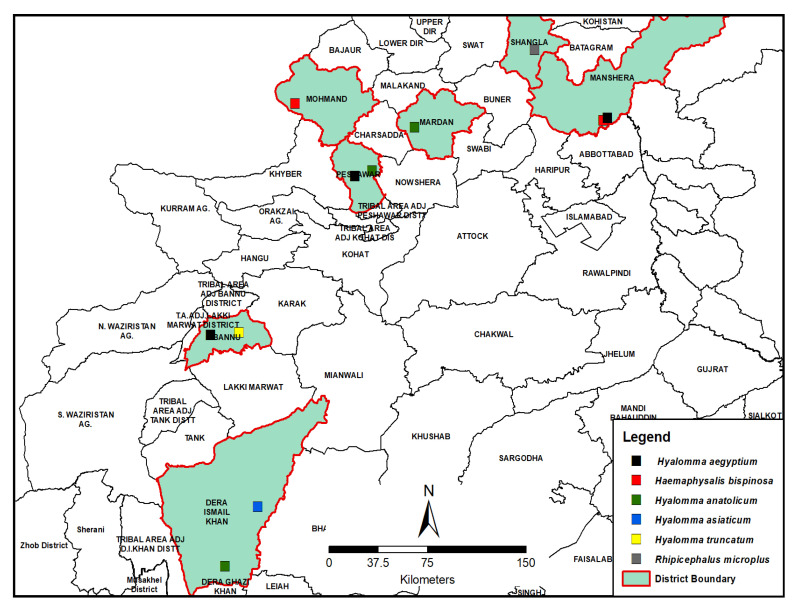
Different sites showing the dominance index of different tick species in KPK, Pakistan.

**Table 1 ijerph-19-11759-t001:** GPS locations of study sites for the collection hard ticks during the study period.

Agro-Climatic Zones	Sites	Sub-Sites	Latitude	Longitude
Sub-humid Eastern Mountains	Mansehra	Dhangri	34°.32′05° N	73°.18′07° E
		Sufaida	34°.32′85° N	73.24′70° E
Higher Northern Mountains	Mingora	Barikot	34°39′59.35 N	72°14′29.01 E
		Kabal	34°46′58.34 N	72°17′34 E
Central Valley Plains	Mardan	Rashakai	34°07′05.58 N	72°01′35.21 E
		Takht Bai	34°16′53.86 N	71°56′38.71 E
	Peshawar	Darmangi	34°02′42.75 N	71°31′57.17 E
		Chankari	33°59′54.14 N	71°39′05.44 E
Suleiman Piedmont	Bannu	Nurar	32°53′51.95″ N	70°32′53.94″ E
		Taji Khula	32°57′3.55″ N	70°44′23.72″ E
	Dera Ismail Khan	Topan Wala	31°49′16.45 N	70°52′11.20 E
		Taji Khula	32°57′3.55″ N	70°44′23.72″ E

**Table 2 ijerph-19-11759-t002:** The overall prevalence of ticks in sheep and goats of KPK province, Pakistan.

Tick Species	Number of Positive Animals	Prevalence (%)
*Rhipicephalus microplus*	542	36.2
*Hyalomma truncatum*	272	18.1
*Hyalomma aegyptium*	169	11.3
*Haemaphysalis bispinosa*	36	2.4
*Hyalomma asiaticum*	103	6.9
*Hyalomma anatolicum*	377	25.2

**Table 3 ijerph-19-11759-t003:** Correlation between different parameters and their effects on prevalence of ticks (n = 1500).

	Weather	Mini. Temp. (C°)	Max. Temp. (C°)	Sunshine (h)	Humidity (%)	Precipitation (mm)	Host Sex	Host Age	Ticks
**Weather**	1.000								
**Mini.Temp.**	−0.036	1.000							
**Max.Temp.**	* 0.069	* 0.873	1.000						
**Sunshine (h)**	* −0.139	* 0.590	* 0.293	1.000					
**Humidity (%)**	* 0.068	* −0.121	* −0.121	* −0.325	1.000				
**Precipitation (mm)**	−0.031	* 0.364	* 0.251	* 0.060	* 0.097	1.000			
**HostSex**	−0.008	0.014	0.026	−0.007	0.024	0.010	1.000		
**Hostage**	* 0.063	−0.015	−0.017	0.009	−0.030	−0.001	0.048	1.000	
**Ticks**	* −0.064	* 0.458	* 0.446	* 0.148	* −0.224	* 0.234	0.019	* −0.104	1.000

* Significant difference at *p* < 0.05.

**Table 4 ijerph-19-11759-t004:** Agro-climatic-zone-wise prevalence of hard ticks in small ruminants of Khyber Pakhtunkhwa (KPK) province, Pakistan.

Tick Species	Agro-Climatic Zones				
	Central Valley Plains	Suleiman Piedmont	Sub-Humid Eastern Mountains	Higher Northern Mountains	Total
*Rhipicephalus microplus*	186 (39.15%)	328 (63.07%)	0	0	514
*Hyalomma anatolicum*	37 (7.78%)	0	270 (100%)	235 (100%)	542
*Hyalomma trancatum*	18 (3.78%)	146 (28.07%)	0	0	164
*Hyalomma asiaticum*	204 (42.94%)	46 (8.84%)	0	0	250
*Haemaphysalis bispinosa*	10 (2.10%)	0	0	0	10
*Hyalomma* *aegyptium*	20 (4.21%)	0	0	0	20
Total	475	520	270	235	1500
	Chi-Square (χ^2^) 4327.4		*p*-value 0.000		

**Table 5 ijerph-19-11759-t005:** Frequency distribution of hard ticks with reference sampling sites, land topography, host gender, and age groups in sheep and goats.

Variables	Levels	Frequency	Percentage (%)
Sampling sites	Barikot	109	7.3
	Chankari	109	7.3
	Darmangi	131	8.7
	Dhangri	147	9.8
	Kabal	126	8.4
	Nurar	100	6.7
	Rashakai	112	7.5
	Shorkot	193	12.9
	Sufaida	123	8.2
	Taji khula	92	6.1
	Takht bai	123	8.2
	Topan wala	135	9.0
	Chi-Square (χ^2^) = 848.100		*p*-value 0.000
Land topography	Hilly	147	9.8
	Semi hilly	785	52.3
	Plain area	568	37.9
	Chi-Square (χ^2^) = 1595.938		*p*-value 0.000.

**Table 6 ijerph-19-11759-t006:** Shannon–Weiner diversity index of tick species.

Hard Ticks	n = Frequency of Ticks	=n/N	Antilog	n/Nx Antilog	H
*Rhipicephalus microplus*	514	0.342667	−1.070	−0.36665	0.36665
*Hyalomma anatolicum*	542	0.361333	−1.017	−0.36748	0.36748
*Hyalomma truncatum*	164	0.109333	−2.213	−0.24195	0.24195
*Hyalomma asiaticum*	250	0.166667	−1.791	−0.2985	0.2985
*Haemaphysalis bispinosa*	10	0.006667	−5.010	−0.0334	0.0334
*Hyalomma aegyptium*	20	0.013333	−4.317	−0.05756	0.05756
Total = N	1500				

**Table 7 ijerph-19-11759-t007:** Diversity and species richness of ticks in the studied areas of Khyber Pakhtunkhwa (KPK) province, Pakistan.

Study Sites	Dominance_D	Shannon_H	Simpson_1-D	Evenness_e^H/S	Equitability_J
**Barikot**	0.04465	3.907	0.9554	0.535	0.862
**Chankari**	0.1069	2.957	0.8931	0.5346	0.8252
**Darmangi**	0.1081	3.036	0.8919	0.5079	0.8176
**Dhangri**	0.08896	3.443	0.911	0.4469	0.8104
**Kabal**	0.1074	2.407	0.8926	0.7928	0.912
**Nurar**	0.06278	2.883	0.9372	0.8505	0.9468
**Rashakai**	0.08419	3.162	0.9158	0.4818	0.8124
**Shorkot**	0.1176	2.868	0.8824	0.5177	0.8133
**Sufaida**	0.08552	3.147	0.9145	0.475	0.8087
**Taji Khula**	0.08626	3.133	0.9137	0.4779	0.8093
**Takht Bai**	0.1251	2.874	0.8749	0.4787	0.796
**Topan Wala**	0.08112	3.203	0.9189	0.5858	0.8569

## Data Availability

Tick specimens have been deposited in the Department of Zoology, Wildlife and Fisheries at the PMAS-Arid Agriculture Rawalpindi and are accessible for study.

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
