# Peer review of "Prevalence and Distribution of Hard Ticks and Their Associated Risk Factors in Sheep and Goats from Four Agro-Climatic Zones of Khyber Pakhtunkhwa (KPK), Pakistan"

_ijerph, 2022, doi:10.3390/ijerph191811759_

Round 1
Reviewer 1 Report
The authors attempted to work out epidemiological profile of hard ticks prevalent in small ruminants being reared in various agro-climatic zones of Khyber Pakhtunkhwa (KPK), Pakistan.
This is a good piece of work and it attempts to find the prevalence of ticks in the different agroecological zones. However, i feel the authors should address the following concerns:-
1. The introduction does not adequately justify the study
2. The authors did not link the tick prevalence to tick borne diseases in these areas or at least the prevalence of tick borne parasites among the collected ticks
3. The authors should have shown the relative importance of the different tick species found among the small ruminants.
Author Response
"Please see the attachment"

Reviewer 2 Report
The manuscript can generally be accepted for publication after some reviews and attending to the following comments
1. Overall observation
The authors highlights that; “Previous surveillance studies on tick diversity were representative of different areas which not truly demonstrating the target areas. In KPK province detailed epidemiological investigation has not been carried out in different agro-climatic zones. However, I found a similar study published in March 2022 in ijerph journal “” Epidemiology, Distribution and Identification of Ticks on Livestock in Pakistan (https://www.mdpi.com/1660-4601/19/5/3024/htm) ””. In my view, this study may look like was from the same area and perhaps more comprehensive, making the current study a subset of this previous one. Can the authors explain whether this study was unique to avoid repetitiveness of already published work? Morover, in this current study, samples were collected between May 2017 to April 2018 almost 5 years ago, compared with the already published study above where samples were collected 2019-2020, making it more recent.
2. Study title
The study title “Epidemiology of hard ticks in Sheep and Goats from various 2 agro-climatic zones of Khyber Pakhtunkhwa (KP), Pakistan. For lack of a lot of considerably epidemiological variations, the authors should keep the title simple. I suggest “Prevalence and distribution of Hard Ticks and Their Associated Risk Factors in Sheep and Goats from various 2 agro-climatic zones of Khyber Pakhtunkhwa (KP), Pakistan.
3. Statistical language in the abstract section
Line 22-31; Simplify the statistical language in the abstract section to make the readers understand your communication by mere reading at the abstract, foristance “”” correlation coefficient values and diversity indices as used in the abstract section should be substituted by interpreted findings, not to send back the reader into statistics to read and find out what exactly you are meaning. Additionally throughout the results section, you present unnecessarily a lot of statistical data than interpreted data, reconsider please
4. Italicizing species names
Line 32-37, tick species names scientific writing order has been ignored. Correct here and elsewhere throughout the manuscript. The same applies to the discussion
Author Response
'Please see the attachment"
